# OpenReview forum: "Distractor Injection Attacks on Large Reasoning Models: Characterization and Defense"
_ICLR.cc/2026/Conference — Submitted to ICLR 2026_

### Official Review · Reviewer_PydC · 2025-10-27

**Soundness:** 3
**Presentation:** 3
**Contribution:** 3
**Rating:** 4
**Confidence:** 3

**Summary:**

The work propose a distractor injection attack for reasoning models that hijacks the reasoning process to make the model deviate from its main task and perform a hidden task, affecting the final output of the model in the context of the primary task. Substantial empirical experiments are conducted across a broad range of distractor tasks, benchmarks, and models, showing consistently that reasoning models are prone to distractions. Furthermore, the work constructs a DPO and SFT dataset based on the attack for adversarial training and show that robustness to the attack can be increased. The work is especially interesting because the threat model incorporates injection attacks to LLM judges, a highly concerning setting that has gained recent attention due to malicious prompts being injected into paper submissions in anticipation of reviews generated by LLMs.

**Strengths:**

The threat model takes the setting of injection attacks that make the target LLM defy the primary task system prompt. As mentioned in the summary, this is a timely topic considering real-world injection attacks within paper submissions targeting possible LLM reviewers. It is surprising to see that a wide range of LRMs fall prey to this attack, which is alarming because reasoning models are gaining more popularity due to better performance. The work provides substantial empirical results to back this claim.

**Weaknesses:**

1. I am confused about the practicality of the threat model outside of an LLM judge setting. When attacking an LLM judge with this attack, the assumption is that the user=attacker $\neq$ judge, where some third-party is using the LLM as a judge. The attacker, hence user, wants to bypass the LLMs main judgement to hack for a higher reward for a certain response. However, when performing this attack on a normal task like MATH, the setting is that the user=attacker is just trying to get a correct answer to a MATH problem from their own LLM. Why would the user attack its own model and try to get a distracted reasoning or wrong answer?
2. Regarding the point above, I would like to see examples of $P_sys$ (the system prompt) for each task (MMLU-Redux, MATH, IFEval, BFCL V3, JudgeLM) to better understand what the goal of the attacker is. This is also because the threat model assumes an attack is successful if $P_sys$ is violated.
3. What is the performance of RPO [1] for adversarial training, which is DPO+SFT simultaneously?
4. Why is Implicit Compliance not as concerning as Covert Compliance, if not more? Implicit Compliance reveals no recognition of distraction in both the reasoning and answer which seems more problematic than recognition in at least the reasoning regardless of exposure to the user.


**References**

[1] Pang, Richard Yuanzhe, et al. "Iterative reasoning preference optimization." Advances in Neural Information Processing Systems 37 (2024): 116617-116637.

**Questions:**

No further comments.

---

### Official Review · Reviewer_9mfL · 2025-11-01

**Soundness:** 2
**Presentation:** 3
**Contribution:** 2
**Rating:** 2
**Confidence:** 3

**Summary:**

This paper proposes a new task called "reasoning distraction", a type of prompt injection. The author claim "structurally distinct subclass that uniquely exploits the chain-of-thought process, rather than issuing direct command-style manipulations." (which I am not completely convinced). In other words, they claim that the goal of existing prompt injection is to "Ignores instructions or produces irrelevant/unsafe content.", but  "reasoning distraction" is to "Solves an injected complex task, abandoning or corrupting the primary task.".

Under the task setting, they design five categories of the Distractor Task, i.e., Mathematical Reasoning, Coding, Logical Reasoning, Symbolic Reasoning and Simple Arithmetic. Next, they evaluate a diverse set of SOTA LRMs. Finally, based on the distraction datasets, they do DPO to improve model's robustness to these distraction, and the performances demonstrate the effectiveness.

**Strengths:**

1. It focuses on LLM reasoning safety issues, and demonstrate it is still alarming and SOTA LLMs are vulnerable to such attacks
2. The analysis is comprehensive and systematic.

**Weaknesses:**

1. The differences of reasoning distraction and regular prompt injection is not clear to me. Or put it in another way, why we need to care about such as unique category of the prompt injection? any different solutions should be considered? is that more serious/difficult to solve?
2. There is no systematic comparison between regular prompt injection and the distraction task setup.
3. More deeper analysis is encouraged, such as  can we detect such  objective diversion in latent space? and any lightweight solution? no training required.

**Questions:**

See weakness above

---

### Meta-Review · Area_Chair_RsMv · 2026-01-06

**Summary:**

This paper studies “distractor injection” attacks on reasoning models, showing that inserting irrelevant but complex tasks can derail step-by-step reasoning. It also proposes a training-based defense using synthetic adversarial data.

**Reviewer Concerns:**

The main concern across reviews is the threat model and clarity of positioning: outside of the LLM-judge scenario, it’s not clear why a “user” would attack their own model on standard tasks, and the paper does not convincingly separate “reasoning distraction” from more general prompt injection (nor provide systematic comparisons/baselines).  Reviewers also asked for clearer task/system-prompt examples and deeper analysis (e.g., detection or lighter-weight defenses).

**Reviewer Scores:**

Reviewer 9mfL: I don’t think they would move much in discussion. Their core concern is that “reasoning distraction” doesn’t feel clearly distinct from standard prompt injection, and they also ask for systematic comparisons / deeper analysis (e.g., detection, lightweight defenses).

Reviewer PydC: This reviewer is already close to the bar and generally likes the topic/empirics, but is hung up on the practicality of the threat model outside the LLM-judge setting and asks for clearer system-prompt examples.

---

### Decision · Program_Chairs · 2026-01-26

Reject